# A Coherent Wideband Acoustic Source Localization Using a Uniform Circular Array

**DOI:** 10.3390/s23115061

**Published:** 2023-05-25

**Authors:** Meng Jiang, Chibuzo Joseph Nnonyelu, Jan Lundgren, Göran Thungström, Mårten Sjöström

**Affiliations:** Sensible Things that Communicate Research Centre, Mid Sweden University, 852 30 Sundsvall, Sweden

**Keywords:** array manifold interpolation, direction of arrival estimation, wideband sources

## Abstract

In modern applications such as robotics, autonomous vehicles, and speaker localization, the computational power for sound source localization applications can be limited when other functionalities get more complex. In such application fields, there is a need to maintain high localization accuracy for several sound sources while reducing computational complexity. The array manifold interpolation (AMI) method applied with the Multiple Signal Classification (MUSIC) algorithm enables sound source localization of multiple sources with high accuracy. However, the computational complexity has so far been relatively high. This paper presents a modified AMI for uniform circular array (UCA) that offers reduced computational complexity compared to the original AMI. The complexity reduction is based on the proposed UCA-specific focusing matrix which eliminates the calculation of the Bessel function. The simulation comparison is done with the existing methods of iMUSIC, the Weighted Squared Test of Orthogonality of Projected Subspaces (WS-TOPS), and the original AMI. The experiment result under different scenarios shows that the proposed algorithm outperforms the original AMI method in terms of estimation accuracy and up to a 30% reduction in computation time. An advantage offered by this proposed method is the ability to implement wideband array processing on low-end microprocessors.

## 1. Introduction

In sound source localization (SSL), also known as acoustic direction of arrival estimation (DoA), the bearing angles (and in some cases range) of sound sources are estimated from the measurements taken by sampling the acoustic waves generated by the sources at different positions and orientation in space. SSL for wideband sources can be very computationally demanding due to the complexity of the existing algorithms. Thus, a low-complexity and computational efficient algorithm will greatly benefit embedded system real-time applications with a low-end microprocessor.

Direction of arrival estimation finds application in speaker identification and localization [1,2], source localization in humanoid robots and drones [3,4], ground vehicle tracking [5], geometrical room modeling [6], and many others. Various direction of arrival estimation algorithms for narrowband sources have been proposed over the years. Some of the well-known algorithms include the Multiple Signal Classification (MUSIC) algorithm and its variants [7,8], the estimation of signal parameters via rotational invariance (ESPRIT) algorithm and its variants [9], minimum variance distortionless response (MVDR) and its variants [10,11], the generalized correlation method [12], and many others [13].

In general, most real-life sound sources are wideband signals; hence, the narrowband assumption adopted for the development of the narrowband beamforming method does not hold. For wideband sources, the phase difference between the outputs of the sensors is not only dependent on the direction of arrival, but also on the temporal frequency. Some methods have been proposed to overcome this challenge. The wideband signal can be seen as a collection of multiple narrowband signals; hence, the narrowband beamforming methods can be applied to each narrowband bin and averaged over the entire wideband. These approaches are referred to as the incoherent signal subspace method (ISSM). Examples of such methods are the incoherent MUSIC (iMUSIC) [14], test of orthogonality of projected subspaces (TOPS) [15], test of orthogonality of frequency subspaces (TOFS) [16], and weighted squared test of orthogonality of projected subspaces (WS-TOPS) [17]. This approach suffers not only from severe degradation when correlated multipath exists but also lacks statistical stability compared to the wideband coherent methods. However, given sufficient signal-to-noise ratio (SNR), the iMUSIC algorithm performs well yielding sharp peaks in the beampattern [18]. It requires a frequency selection step as the inclusion of low-SNR frequency bins tends to degrade the resulting beampattern, and thereby reducing the source peak and introducing spurious peaks.

An alternative approach to wideband problems is the coherent method. In this approach, the covariance matrix of the component narrowbands is coherently summed into one narrowband covariance matrix, whereby the narrowband subspace method is applied to the universal covariance matrix. This is achieved by multiplying each narrowband cross-covariance matrix by a focusing matrix and then summed to obtain the steered covariance matrix. The earliest of this approach was proposed as the Coherent Signal-Subspace (CSS) method [19]. Other coherent signal-subspace methods have since been proposed: the least square and total least-square coherent signal subspace (CSS-MTLS) [20], Particle Swarm Optimization algorithm (PSO) [21], Dir-MUSIC algorithm using the strength proportion (DirSP) [22], and array manifold interpolation (AMI) [23]. A major advantage of the CSS methods is their ability to handle correlated sources. On the other hand, the performance of the CSS is sensitive to the initial values of the directions of arrival. This is because the focusing matrix of the CSS method is dependent on both directions of arrival and narrowband frequencies, making the initial estimation of the directions of arrivals an intermediate step.

The array manifold interpolation (AMI) method [23] is made possible by the series expansion of the plane wave. This gives a focusing matrix that does not depend on the initial direction of arrival of the incident signals, but only on the narrowband frequencies [18,23]. This leads to a less computational CSS method for wideband sources. The complexity is further reduced for the uniform circular array as the sensor spacings are identical [18]. However, AMI depends on the Bessel function of the first kind, which may make the implementation of this algorithm on embedded systems computationally inefficient.

Following the advances in the fields of embedded systems and Internet-of-Things (IoT), SSL and DoA methods are now being implemented on embedded systems. They are sometimes implemented together with other functionality, competing over the computational resources available on the embedded systems. Recent research in the fields of SSL and DoA has been conducted, where computational complexity is in focus, both on a model development level [24] and in more implementation on application papers [25]. These are, however, targeting narrowband signals, which cannot be adopted—even with slight modification—for wideband sources.

In this paper, a modified array manifold interpolation method is proposed for the uniform circular array of omnidirectional microphones. The proposed method takes advantage of the Bessel function’s asymptotic property and forms a UCA-specific focusing matrix. Combining the assumption of low-frequency wideband signals with an adequate signal-to-noise ratio, the proposed focusing matrix can be approximated as the product of a discrete Fourier transform (DFT) matrix and a diagonal matrix. The entries of the diagonal matrix are the *n*th power of the ratio of the central processing frequency and the narrowband frequencies. Thus, the computational complexity reduction can be done by eliminating the calculation of the Bessel function. The simulation and experimentation results of the proposed method are compared with iMUSIC, WS-TOPS, and the original AMI method due to these particular existing methods do not require an initial estimate of the DoA [26]. The proposed modification reduces the computational complexity required for generating the focusing matrix, and yields a similar direction of arrival estimation performance to the original AMI method.

The rest of this paper is organized as follows: the signal model and array manifold interpolation are described in Section 2, the proposed algorithm is developed in Section 3, its efficacy is shown by Monte Carlo simulation in Section 4, and the laboratory experiment reported in Section 5. Finally, the paper is concluded in Section 6.

## 2. Array Manifold Interpolation

The signal model is described in Section 2.1 where the received data and array manifold are described. The steered covariance matrix via array manifold interpolation is introduced in Section 2.2 and applied to the uniform circular array in Section 3.1.

### 2.1. Signal Model

Considering *L* (either known a priori or estimated) number of wideband sources incident on a uniform circular array of radius *r* with *M* omnidirectional microphones, as shown in Figure 1, which illustrates the *ℓ*th incident source. The bandwidth of the sources are not necessarily identical but must exist within some overlap bounded by $\left[f_{\min},f_{\max}\right]$. The output of the *m*th sensor is given as:(1)$$\begin{array}{ccc} x_{m}\left(t\right) & = & \sum_{\ell =1}^{L}s_{\ell }\left(t-\tau_{\ell m}\right)+v_{m}\left(t\right) \end{array}$$
where $s_{\ell }\left(\cdot \right)$ is the *ℓ*th incident source, $\tau_{\ell m}=k_{\ell }^{T}r_{m}/\omega$ is the *ℓ*th signal’s time-delay-of-arrival for the *m*th sensor and the reference point (center of the circular array), ω=2πf is the source angular frequency, $k_{\ell }=\left(2\pi /\lambda \right){\left[\cos\left(\phi_{\ell }\right)\sin\left(\theta_{\ell }\right)\;\sin\left(\phi_{\ell }\right)\sin\left(\theta_{\ell }\right)\;\cos\left(\theta_{\ell }\right)\right]}^{T}$ is the wavenumber of the *ℓ*th source, and $r_{m}$ is the position vector of the *m*th sensor. Furthermore, in (Equation 1), $v_{m}\left(t\right)$ is the additive noise on the *m*th sensor, assumed to be zero-mean spatio-temporally uncorrelated Gaussian noise with a priori known variance. Assuming the source and array lie on the same plane (where $\theta_{\ell }=90^{\circ }$), for the uniform circular array, the time delay is $\tau_{\ell m}=kr_{m}\cos\left(\phi_{\ell }-\psi_{m}\right)/\omega$, where $\phi_{\ell }\in \left[0,2\pi \right)$ is the azimuth angle of incidence of the *ℓ*th source and $\psi_{m}=2\pi m/M$ is the angle between the line from the origin to the *m*th sensor and the positive *x*-axis. The wavenumber magnitude $k=|k_{\ell }|=2\pi f/c_{\mathrm{air}}$, where $f\in [f_{\min},f_{\max}]$ is the source-frequency, and $c_{\mathrm{air}}=343\;{\mathrm{ms}}^{-1}$ is the speed of sound propagation in dry air. With an adequate number of observations, the *j*th sample of the discrete Fourier transform (DFT) of the *m*th sensor’s output:(2)$$\begin{array}{ccc} X_{m}\left(k_{j}\right) & = & A\left(\phi ,k_{j}\right)S\left(k_{j}\right)+V_{m}\left(k_{j}\right),\;\;\;j=1,2,\cdots ,J \end{array}$$
where *J* is the total number of frequency samples, $A\left(\phi ,k_{j}\right):=\left[a\left(\phi_{1},k_{j}\right)\;a\left(\phi_{2},k_{j}\right)\;\cdots \;a\left(\phi_{L},k_{j}\right)\right]$ is the steering matrix, $S\left(k_{j}\right)=[S_{1}\left(k_{j}\right)\;S_{2}\left(k_{j}\right)\;$$\cdots \;S_{L}\left(k_{j}\right){]}^{T}$ is the DFT of the incident signals, $V_{m}\left(k_{j}\right)$ is the DFT of the additive noise, and:(3)$$\begin{array}{ccc} a(\phi_{\ell },k_{j}) & = & {\left[e^{ik_{j}r\cos\left(\phi_{\ell }-\psi_{1}\right)}\;e^{ik_{j}r\cos\left(\phi_{\ell }-\psi_{2}\right)}\;\cdots \;e^{ik_{j}r\cos\left(\phi_{\ell }-\psi_{M}\right)}\right]}^{T} \end{array}$$
is the M×1 array manifold at wavenumber $k_{j}$. The variable *i* denotes the imaginary unit, i.e., $i=\sqrt{-1}$.

The correlation matrix $E[X\left(k_{j}\right)X{\left(k_{j}\right)}^{H}]$ of the received data at a given wavenumber $k_{j}$ is estimated as:(4)$$\begin{array}{ccc} R_{j} & = & \frac{1}{Q}\sum_{q=1}^{Q}X^{\left(q\right)}\left(k_{j}\right)X^{\left(q\right)}{\left(k_{j}\right)}^{H} \end{array}$$
where *Q* is the number of independent measurements and $X^{\left(q\right)}\left(k_{j}\right)=[X_{1}^{\left(q\right)}\left(k_{j}\right)\;X_{2}^{\left(q\right)}\left(k_{j}\right)\;\cdots$$\;X_{M}^{\left(q\right)}\left(k_{j}\right)]$ is the discrete Fourier transform of the *q*th measurement. Superscript ${\left(\cdot \right)}^{H}$ denotes the conjugate transpose.

### 2.2. The Array Manifold Interpolation Method

The correlation matrix of the observation has been developed for narrowband frequency $f_{j}$. For wideband sources, the various narrowbands are coherently summed into one correlation matrix by multiplying each narrowband correlation matrix by a focusing matrix. One method of achieving this is the array manifold interpolation method (AMI) [23]. This method is introduced in this section for an arbitrary array geometry and then further simplified for the uniform circular array in the next section.

The *m*th element of the array manifold of the planar array is represented as a series expansion of a plane wave, i.e., the Jacobi–Anger expansion as:(5)$$\begin{array}{ccc} {\left[a\left(\phi ,k\right)\right]}_{m} & = & e^{ikr_{m}\cos\left(\phi -\psi_{m}\right)}\\ & = & \sum_{n=-\infty }^{\infty }i^{n}J_{n}\left(kr_{m}\right)e^{-in\psi_{m}}e^{in\phi }\\ & = & {\tilde{b}}_{m}\left(k\right)\tilde{w}\left(\phi \right) \end{array}$$
where ${\left[{\tilde{b}}_{m}\left(k\right)\right]}_{n}:=i^{n}J_{n}\left(kr_{m}\right)e^{-in\psi_{m}}$ is *n*th element of the row vector ${\tilde{b}}_{m}\left(k\right)$, and ${\left[\tilde{w}\left(k\right)\right]}_{n}:=e^{in\phi }$ is the *n*th element the column vector $\tilde{w}\left(\phi \right)$, for n=−∞,⋯,0,⋯,∞. The function $J_{n}\left(\cdot \right)$ is the Bessel function of the first kind with order *n*, $r_{m}$ is the distance between the origin (usually the center of moment of the geometry) and the *m*th microphone, and other variables are as previously defined.

The Bessel function decays exponentially as $n\gg kr_{m}$; hence, the infinite summation in (6) can be truncated to a given *N* such that $|J_{n}\left(k_{\max}r_{m,\max}\right)|<\epsilon$ for n>N, where ϵ is an acceptable truncation error. This implies that the length of vectors ${\tilde{b}}_{m}\left(k\right)$ and $\tilde{w}\left(\phi \right)$ is limited to 2N+1, thus reducing to $b_{m}\left(k\right)$ and w(ϕ), respectively. It is practical to set $N\geq 2k_{\max}r_{m,\max}$. Generally, the array manifold of the array with *M* total number of microphones is given by:(6)$$\begin{array}{ccc} a(\phi ,k) & = & B(k)w(\phi ) \end{array}$$
where ${\left[B\left(k\right)\right]}_{m,n}=i^{n}J_{n}\left(kr_{m}\right)e^{-in\psi_{m}}$, m=1,2,⋯,M, and n=−N,−(N−1),⋯,0,⋯,N−1,N and w(ϕ) is as previously defined but for n=−N,−(N−1),⋯,0,⋯,N−1,N.

For wideband sources, the concept of frequency smoothing is adopted to coherently sum the component narrowband covariance matrices of the wideband source into one narrowband covariance matrix. This is followed by the smoothing of the mapped narrowband covariance matrix. For this, we define an M×M focusing matrix $T(k_{j})$ such that:(7)$$\begin{array}{ccc} T\left(k_{j}\right)a\left(\phi ,k_{j}\right) & = & a(\phi ,k_{0}),\;\;\forall \phi . \end{array}$$
According to (Equation 6) and (Equation 7):(8)$$\begin{array}{ccc} T(k_{j}) & = & B\left(k_{0}\right)B{\left(k_{j}\right)}^{\#} \end{array}$$
where $B{\left(k_{j}\right)}^{\#}={\left(B^{H}B\right)}^{-1}B^{H}$ is the generalized inverse of matrix $B(k_{j})$ and $k_{0}\in \left(k_{\min},k_{\max}\right)$ is the central processing wave number. It is becomes necessary to choose *N* and *M* such that M=2N+1 to ensure $B(k_{j})$ is nonsingular [18,23].

The focused and frequency-smoothed covariance matrix is evaluated as:(9)$$\begin{array}{ccc} \tilde{R} & = & \sum_{j=1}^{J}T\left(k_{j}\right)R_{j}T{\left(k_{j}\right)}^{H} \end{array}$$
where *J* is the total number of selected frequency samples as in (Equation 2) and $R_{j}$ is as previously defined in (Equation 4). In order to guarantee an improved performance, the frequency samples in the signal bandwidth that corresponds to the *J* highest DFT magnitudes are selected to avoid low-SNR frequency samples.

## 3. The Proposed Method

### 3.1. For the Uniform Circular Array

For the uniform circular array, $r_{m}=r\;\forall m$ and $\psi_{m}=2\pi m/M$. Thus, the matrix B(k) as defined in (Equation 6) can be expressed as:(10)$$\begin{array}{ccc} B(k) & = & \sqrt{M}FJ\left(k\right) \end{array}$$
where F is a M×(2N+1) unitary DFT matrix defined as:(11)$$\begin{array}{ccc} {\left[F\right]}_{m,n} & = & \frac{1}{\sqrt{M}}e^{-i\frac{2\pi }{M}mn},\;\;\;\begin{array}{c} m=0,1,\cdots ,M-1\\ n=-N,\cdots ,N \end{array} \end{array}$$
and $J\left(k\right)\in C^{\left(2N+1)\times (2N+1\right)}$ is a diagonal matrix with entries:(12)$$\begin{array}{ccc} {\left[J\left(k\right)\right]}_{n,n} & = & i^{n}J_{n}\left(kr\right). \end{array}$$

By setting M=2N+1, F becomes an invertible square matrix. Thus, the focusing matrix is expressed as:(13)$$\begin{array}{ccc} T(k_{j}) & = & FD\left(k_{j}\right)F^{-1}\\ & = & FD\left(k_{j}\right)F^{H} \end{array}$$
since F is a unitary matrix. Matrix D(k) is a (2N+1)×(2N+1) diagonal matrix with diagonal entries:(14)$$\begin{array}{c} {\left[D\left(k_{j}\right)\right]}_{n,n}=\frac{J_{n}\left(k_{0}r\right)}{J_{n}\left(k_{j}r\right)}. \end{array}$$

### 3.2. The Proposed Focusing Matrix

In this section, the original array manifold interpolation method has been introduced and further simplified for the uniform circular array. For an embedded implementation of such an algorithm, a high-end microcontroller capable of implementing the Bessel function evaluation will be required. This requirement translates to higher cost for some low-budget applications. To eliminate the complexity required for the calculation of the Bessel function, the proposed algorithm exploits a property of the Bessel function, which allowed for the total elimination of its computation.

According to the asymptotic property of the Bessel function, for orders *n* far greater than the argument, the Bessel function can be approximated as $J_{n}\left(kr\right)\approx {\tilde{J}}_{n}\left(kr\right)$, where:(15)$$\begin{array}{ccc} {\tilde{J}}_{n}\left(kr\right) & := & \left\{\begin{array}{cc} \frac{1}{\Gamma (n+1)}{\left(\frac{kr}{2}\right)}^{n}\; & \mathrm{for}\;n\geq 0\\ \;\\ \frac{{\left(-1\right)}^{n}}{(-n)!}{\left(\frac{2}{kr}\right)}^{n}\; & \mathrm{for}\;n<0 \end{array}\right. \end{array}$$
given that $0<kr<\sqrt{n+1}$, ∀n>0, assuming the least order of n=0 [27] (p. 364). This small-argument approximation makes it possible to further simplify (Equation 14) to:(16)$$\begin{array}{ccc} {\left[\tilde{D}\left(k_{j}\right)\right]}_{n,n} & = & {\left(\frac{k_{0}}{k_{j}}\right)}^{n}\\ & = & {\left(\frac{f_{0}}{f_{j}}\right)}^{n} \end{array}$$
for n=−N,⋯,0,⋯,N, given that 0<kr<1. This condition is obtainable for a combination of source frequency and small radius uniform circular array. The diagonal matrix $D(k_{j})$ simply reduces to a matrix $\tilde{D}\left(k_{j}\right)$ of *n*th power of the ratio of the central processing frequency $f_{0}$ and the narrow band frequency $f_{j}$, and thus, eliminating the computation of the Bessel function of the first kind, and the focusing matrix in (Equation 13) then becomes:(17)$$\begin{array}{ccc} \tilde{T}\left(k_{j}\right) & = & F\;\tilde{D}\left(k_{j}\right)\;F^{H} \end{array}$$

Finally, the steered covariance matrix is calculated using (Equation 9). Though this method adds additional matrix multiplication compared to the CSS method [19], it eliminates the initial estimation of the directions of arrival, which includes several matrix multiplications depending on the algorithm used. This focusing matrix (compared to other wideband methods) could be useful in tracking moving sound source as one focusing matrix is needed if the frequency spectrum of the source remains fairly constant. Furthermore, the proposed focusing matrix also does not depend on the radius of the uniform circular array, making it suitable for any circular array of any radius given that 0<kr<1 is fulfilled.

### 3.3. Approximation Error

The proposed algorithm is based on the small-argument approximation of the Bessel function of the first kind as given in (Equation 15). However, the Bessel function occurs in ratio as in (Equation 14) for the uniform circular array. In this section, the resultant approximation error on matrix $D(k_{j})$ is analyzed. Towards this, the approximated Bessel function is equal to the true value plus an approximation error. That is:(18)$$\begin{array}{ccc} {\tilde{J}}_{n}\left(kr\right) & = & J_{n}\left(kr\right)\pm \epsilon ,\;\;\epsilon >0 \end{array}$$
where ϵ is the approximation error and expressed as:(19)$$\begin{array}{ccc} \epsilon & = & \sum_{m=1}^{\infty }\frac{{\left(-1\right)}^{m+1}}{m!\Gamma (m+n+1)}{\left(\frac{kr}{2}\right)}^{2m+n}. \end{array}$$
For the ± sign in (Equation 18), the ‘+’ sign applies for n≥0. When n<0, ‘+’ sign applies for even *n*, and ‘−’ sign applies for odd *n*, respectively. The ratio in (Equation 16) becomes:(20)$$\begin{array}{ccc} {\left[\tilde{D}\left(k_{j}\right)\right]}_{n,n} & = & \frac{J_{n}\left(k_{0}r\right)\pm \epsilon_{0}}{J_{n}\left(k_{j}r\right)\pm \epsilon_{j}} \end{array}$$
and the approximation error ΔD of matrix $D(k_{j})$ is thus:(21)$$\begin{array}{ccc} \Delta D & = & {\left[D{\left(k_{j}\right)}_{n_{n}}-\tilde{D}\left(k_{j}\right)\right]}_{n,n}\\ & = & \frac{\pm \epsilon_{j}J_{n}\left(k_{0}\right)\pm \epsilon_{0}J_{n}\left(k_{j}\right)}{J_{n}\left(k_{j}\right)\left[J_{n}\left(k_{j}\right)\pm \epsilon_{j}\right]} \end{array}$$
which shows that the approximation error increases as $|k_{0}-k_{j}|$ increases. This implies that choosing $k_{0}$ as the median of the selected high-SNR *J* bins will reduce the approximation error. As shown in Figure 2, a plot of |ΔD| versus *n* and $k_{j}r\in \left[0.2,1\right]$ and $k_{0}r=0.7$, which shows that the least error occur about $k_{j}r=0.7$, where $|k_{0}-k_{j}|=0$.

### 3.4. Computational Complexity

The computational complexity of the proposed method is compared against those of iMUSIC, WS-TOPS, AMI, and proposed method. Recall that *M* is the number of sensors, *J* is the number of narrow band frequency bins ($J^{\prime }<J$ is the total number of selected processing frequency bins for the WS-TOPS method), *Q* is the number of snapshots, and *L* is the number of sources. Table 1 shows the matrix multiplication complexities, while Table 2 shows the matrix decomposition complexities for the methods under comparison.

The computational complexity of computing the Bessel function of the first kind $J_{n}\left(kr\right)$ with accuracy $2^{-v}$ at the rational point is $O(v\;log^{3}vs.log\;log\;v)$ [28]. The evaluation of $k^{n}$(a ζ-digit number) requires O(lognM(ζ)), where M(ζ) is the time complexity of the multiplication algorithm. For the Harvey–Hoeven algorithm, M(ζ)=O(nlogn), which results to $O(n\;log^{2}\;n)$ [29]. The computational complexity in the procposed methods is further reduced by defining the focusing matrix $T(k_{j})$ as:(22)$$\begin{array}{ccc} {\left[T\left(k_{j}\right)\right]}_{m_{r},m_{c}} & = & \sum_{n=-N}^{N}{\left(\frac{f_{0}}{f_{j}}\right)}^{n}W^{n(m_{r}-m_{c})} \end{array}$$
where $m_{r},m_{c}=1,2,\ldots M$ and $W:=e^{i\frac{2\pi }{M}}$. This eliminates the $O(2JM^{3})$ complexity for the matrix multiplication in (Equation 13) for $T(k_{j})$. Plots of the matrix multiplication complexity versus number of sensors and versus number of narrowband frequency bins are shown in Figure 3a and Figure 3b, respectively.

It is observed that the matrix multiplication complexity involved in the iMUSIC is always lower. However, the matrix multiplication complexity of the proposed method is always less than the original AMI and the WS-TOPS.

### 3.5. Direction of Arrival Estimation

For the uniform circular array, a new focusing matrix for coherently summing the narrowband covariance matrices has been proposed in Section 3. Using the steered covariance matrix, any narrowband subspace direction of arrival method or maximum likelihood estimator can be adopted. In this paper, the MUSIC [7] algorithm is applied as the narrowband subspace method to the focused covariance matrix. The direction of arrival is, thus, the direction corresponding to the peaks of the beampattern:(23)$$\begin{array}{ccc} Y(\phi ) & = & {\left[a{\left(\phi ,f_{0}\right)}^{H}U_{N}U_{N}^{H}a\left(\phi ,f_{0}\right)\right]}^{-1}, \end{array}$$
where $U_{N}$ is the noise subspace of $\tilde{R}$ for proposed method. For higher SNR, the number of sources can easily be obtained by detecting a sharp decrease in the ordered eigenvalues of $\tilde{R}$. Another alternative is to assume that the number of sources is equal to the number of sensors minus 1, i.e., M−1, since that is the highest number of sources the array can detect. Furthermore, for the MUSIC algorithm, overestimating is not as detrimental as underestimating the number of sources. The estimation can then be corrected by iteration, where the number of prominent peaks given by the initial assumption is adopted as the number of sources until a finer beampattern is obtained. Other algorithms for estimating the number of source can be adopted [30,31]. The summary of the proposed algorithm is captured in the block diagram in Figure 4.

## 4. Monte Carlo Simulations to Test the Efficacy of the Proposed Method

The efficacy of the proposed algorithm is evaluated in this section. For this purpose, the performance of the proposed method is compared against that of the iMUSIC, WS-TOPS, and the Original AMI. The choice of these methods for comparison is mainly because they do not require initial estimation of the direction of arrival.

### 4.1. Root-Mean-Square Performance

The incident signal is modeled as band-limited real-valued zero-mean Gaussian random variables sampled at a sampling frequency of 12 kHz at various signal-to-noise ratios and 5 snapshots of 8192 times samples each. This chosen frequency band guarantees that the maximum kr is less than 1. The array is a uniform circular array consisting of M=6 omnidirectional microphones with radius r=4.5cm. For the signal bandwidth (f∈[400,1213]Hz), 100 narrowband frequencies (obtained using fast Fourier transform) were used. The root-mean-square error of the proposed algorithm is compared against that of iMUSIC, WS-TOPS, and original AMI method for a single source located at $120^{\circ }$ with different SNR ranging from −10 dB to 40 dB. Figure 5 shows the plot of the root mean square estimation error versus the SNR for the methods under comparison. Each point in Figure 5 and Figure 6 represents the root-mean-square error value of 100 iterations. The proposed method slightly outperforms the original AMI method for low-frequency wideband sources.

To ascertain the relative performance of the proposed method for low frequency signals, the bandwidth of the signal is limited to f∈[166,500]Hz. The root-mean-squared error versus SNR is shown in Figure 6. This shows that the performance of the proposed method is consistent with the original AMI. Furthermore, both outperform the WS-TOPS for low-frequency wideband sources.

### 4.2. Computational Efficiency of the Proposed Algorithm

The difference in the computational complexity of the proposed algorithm and original AMI method has been discussed in Section 3.4. The computational time interpretation of this improvement is studied in this section. The proposed algorithm saves processing time by cutting the time taken to generate the diagonal matrix $D(k_{j})$ by up to 30% on average using MATLAB R2020b on a Core i7 computer. The reduction in computation time is expected to increase when used on lower processors and in embedded microcontroller systems. This is expected as the elimination of the Bessel function for each order *n* reduces the number of calculations by the processor.

## 5. Experimental Evaluation

The MiniDSP UMA-8 mic array was used for the experiment. As shown in Figure 7, it is a circular board of 4.5 cm radius, which has seven onboard omnidirectional MEMS microphones in total: six uniformly placed around the circumference and one at the center [32]. For the purpose of this experiment, only the six circumferential microphones were used to match the uniform circular array described for this study. The incident sound source is white noise located at an azimuth angle $\phi =120^{\circ }$, recorded at a sampling rate of 48 kHz, playing through a loudspeaker.

The experiments demonstrate how the proposed algorithm performs across signal-to-noise ratio, various wideband frequency ranges (i.e., $kr\in [kr_{min},kr_{max}]$), and various broadband sizes. Noise is added at varying SNR from −20 dB to 40 dB with a step of 10 dB. f∈[200,600] is the filtering range for the experiments.

### 5.1. Root-Mean-Square Error Performance versus Signal-to-Noise Ratio

Additive white Gaussian noise of various SNR was added to the recorded signal and the direction of arrival estimated. For each value of SNR, 100 independent estimations were carried out and the root-mean-square error (RMSE) was taken. Figure 8 shows the plot of the RMSE versus SNR for both the proposed and the original AMI method, N=2, and the frequency range was divided into 30 bins. As shown in Figure 8, the proposed algorithm outperformed the original AMI method.

### 5.2. Performance by Central Processing Frequency

In this section, the performance of the proposed algorithm across kr is studied. To ensure every other parameter is the same but the kr, the magnitude of the frequency bins is shifted across the frequency. By doing so, only the $kr_{\min}$, $k_{0}r$, and $kr_{\max}$ vary. For the recorded signal used, the DFT magnitudes for f∈[50,460] (the dominant bins) were shifted in steps of 100 Hz. The central frequency for each case is the median of the selected bins. The RMSE performance versus the central processing frequency $f_{0}$ (or $k_{0}r$) is plotted in Figure 9, which shows that the proposed algorithm performs similarly to the original AMI.

### 5.3. Performance for Different Types of Sources

The proposed algorithm is further tested for various sources. For the experiment, the following sound sources were used: car engine, water fall, hovering drone, electric fan, and train engine. For each sound, the source is moved to four locations, and the direction of arrival estimated using the original array manifold interpolation and the proposed modified method. The average estimation errors for each source type are reported in Table 3 where the bold entries highlights the minimum value in each row.

The performance of the proposed algorithm is shown to outperform the original AMI for all the sound types but the train engine, as shown in Table 3. This goes to show that the proposed method performs similarly to the original AMI, while offering a significant reduction in computational complexity.

## 6. Conclusions

For the uniform circular array, the steered covariance matrix method based on array manifold interpolation is more computationally efficient compared to other coherent covariance matrix averaging. This is due to the fact that the steering matrix in the array manifold interpolation method is independent of the source direction of arrival. It has been shown in this paper that by applying the small-argument approximation of the Bessel function, the *n*th power of the ratio of the Bessel function of different orders are reduced to the *n*th power of the ratio of the processing frequency and the narrowband frequencies. This led to a significant reduction in the computation time for the diagonal matrix component of the focusing matrix, while achieving similar localization accuracy for low-frequency wideband sources compared to the original AMI method. This simplified algorithm eliminates the need for high-performance microprocessors for the implementation of real-time low-frequency wideband sound source localization on the embedded system. Furthermore, the proposed focusing matrix is independent of the array radius, making the focusing matrix reusable for the frequency bins if the signal spectrum remains fairly constant over time.

## Figures and Tables

**Figure 1 sensors-23-05061-f001:**
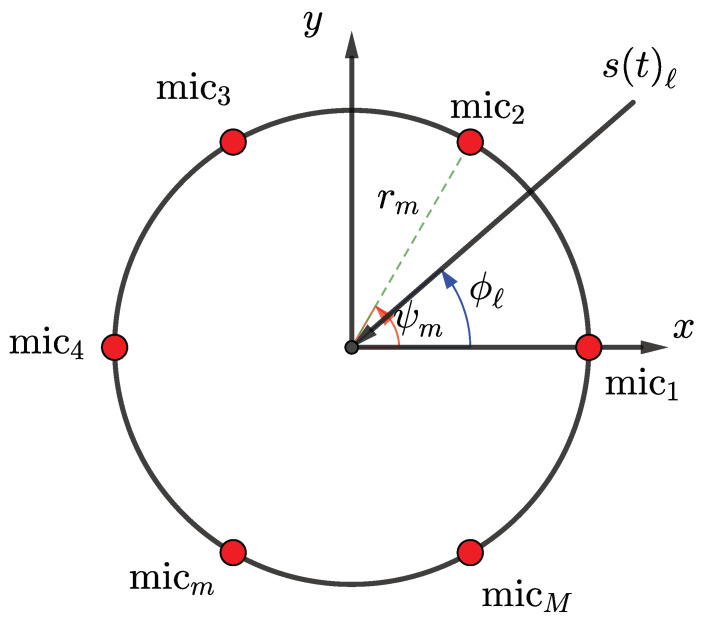
The uniform circular array of omnidirectional microphones with an incident source impinging at azimuth angle $\phi_{\ell }$.

**Figure 2 sensors-23-05061-f002:**
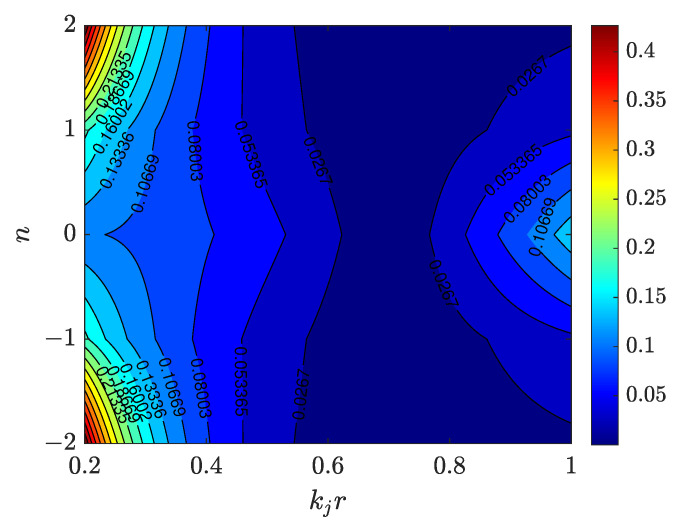
Plot of approximation error ΔD versus order *n* and $k_{j}r\in \left[0.2,1\right]$ for $k_{0}r=0.7$, showing the approximation error increases as the separation distance between processing frequency and other bins increases.

**Figure 3 sensors-23-05061-f003:**
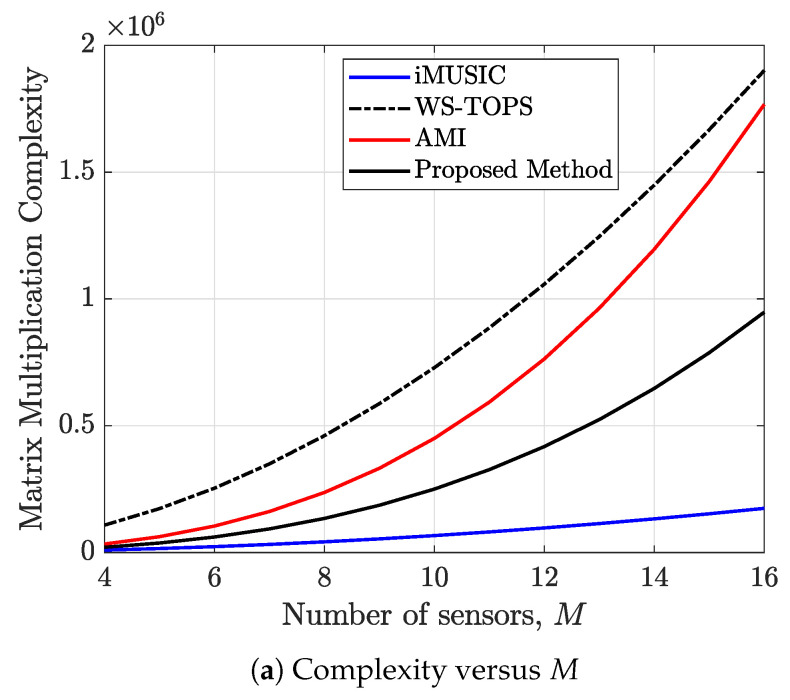
Plot of the matrix multiplication complexity versus (**a**) number of sensors *M* for J=100 bins, L=2, Q=5, and $J^{\prime }=0.1J$, (**b**) number of frequency bins *J* for M=6, Q=5, L=2.

**Figure 4 sensors-23-05061-f004:**
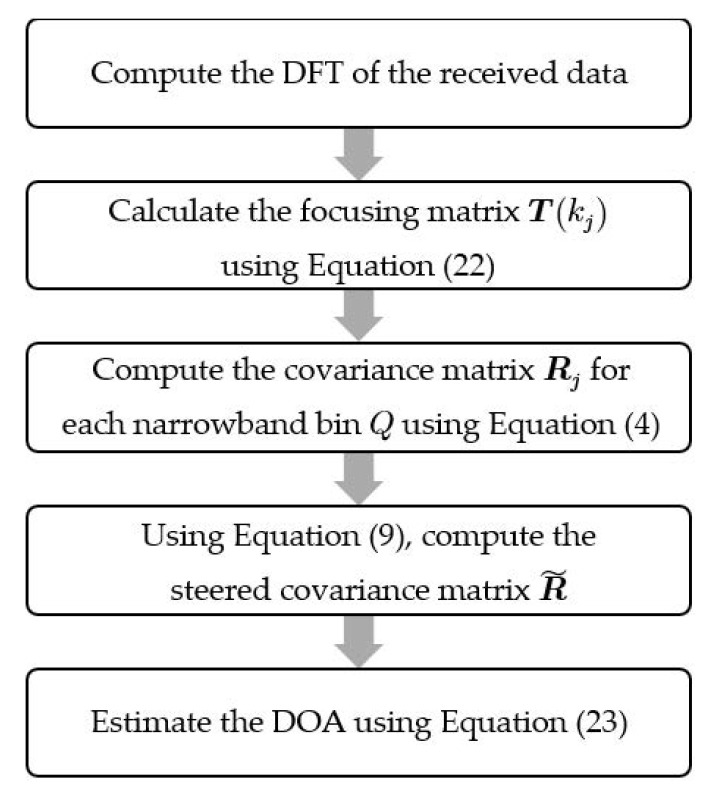
Block diagram showing the summary of the proposed method.

**Figure 5 sensors-23-05061-f005:**
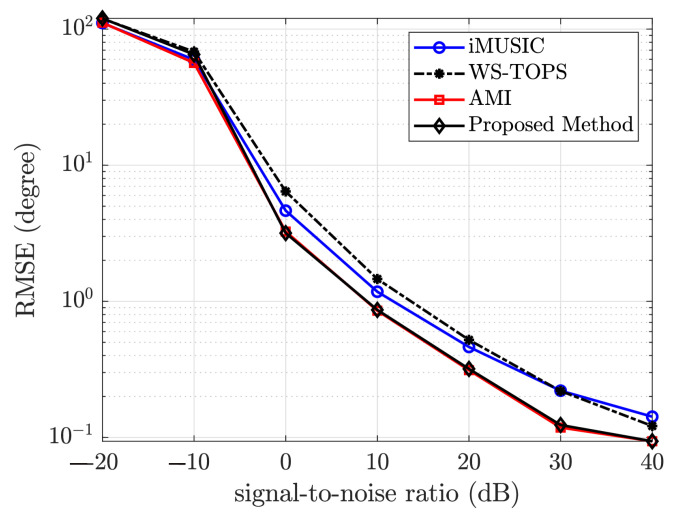
Plot root-mean-squared error versus signal-to-noise ratio for the iMUSIC, WS-TOPS, Original AMI, and proposed algorithm using simulated data.

**Figure 6 sensors-23-05061-f006:**
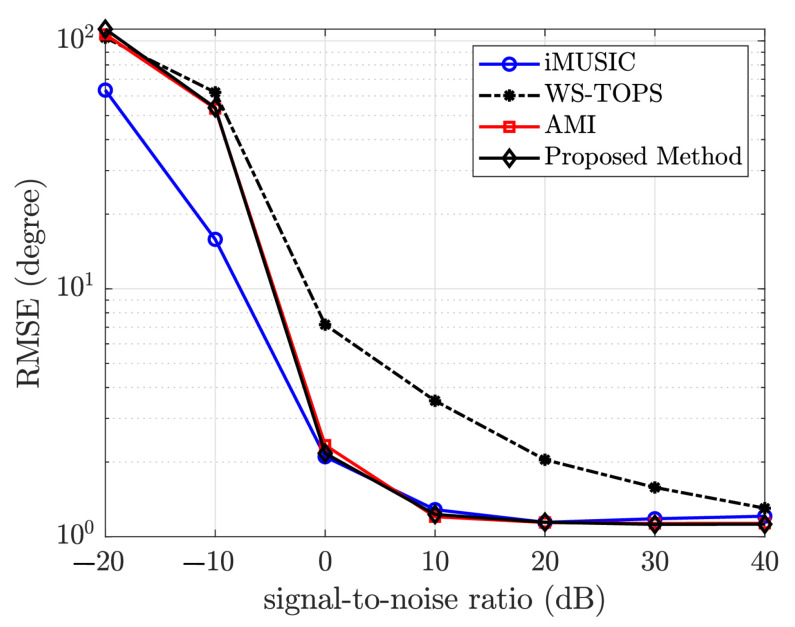
Plot root-mean-squared error versus signal-to-noise ratio for the iMUSIC, WS-TOPS, Original AMI, and proposed algorithm using simulated data of f∈[166,500]Hz.

**Figure 7 sensors-23-05061-f007:**
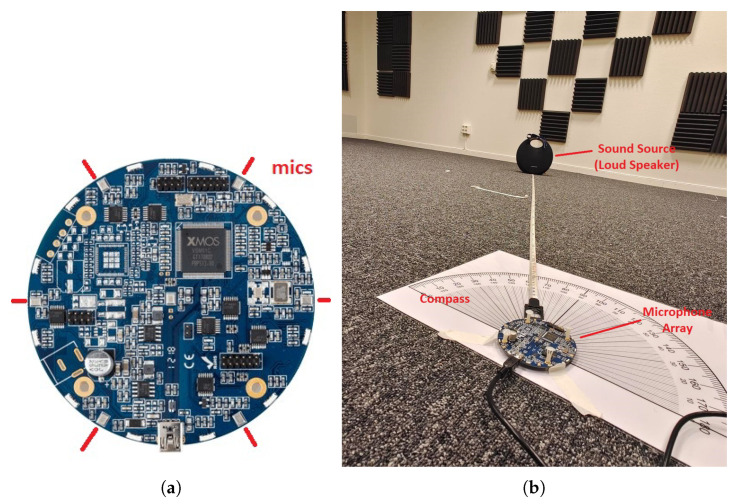
The uniform circular array and the experimental setup. (**a**) The uniform circular array using a MiniDSP UMA-8 system [32] and (**b**) the experimental setup and environment, using sounds from a Harman Kardon Onyx 5 wireless loudspeaker, while measuring distance and angle from the loudspeaker to the array system.

**Figure 8 sensors-23-05061-f008:**
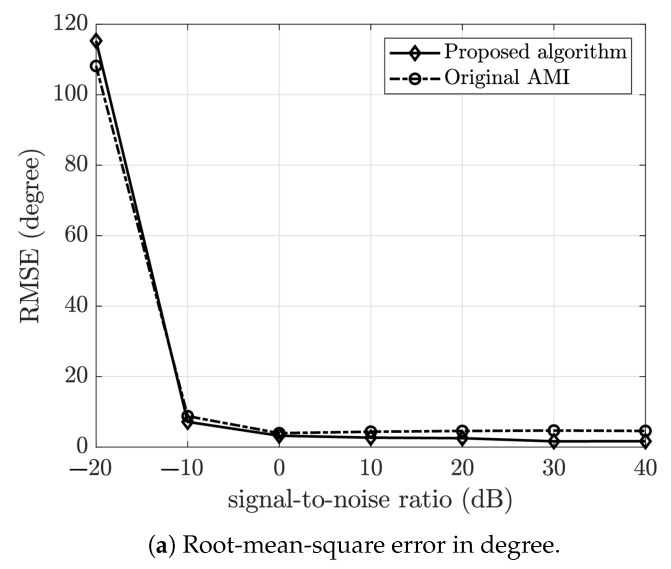
Plot of root-mean-squared error versus signal-to-noise ratio for the proposed algorithm and original AMI method using recorded data.

**Figure 9 sensors-23-05061-f009:**
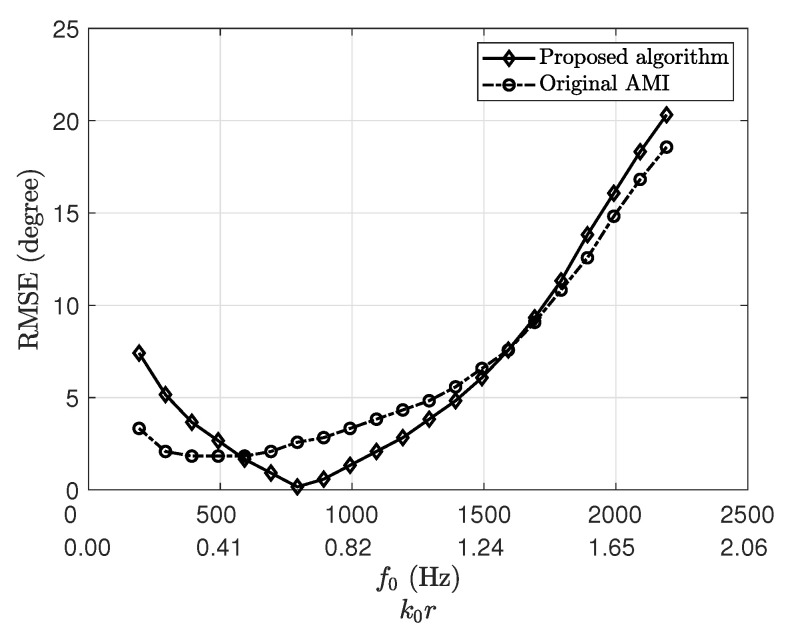
Plot of root-mean-squared error versus central processing frequency $f_{0}$ ($k_{0}r$) for the proposed algorithm and original AMI method using recorded data.

**Table 1 sensors-23-05061-t001:** Matrix multiplication complexity comparison.

Method	Matrix Multiplication
iMUSIC	$O(J\left(M-L\right)\left(2M+1\right)+JQM^{2})$
WS-TOPS	$O(J^{\prime }\left(J-J^{\prime }\right)\left[2LM\left(M-L\right)+L^{2}M+2LM^{2}\right]+\left(J-J^{\prime }\right)QM^{2})$
AMI	$O(\left(M-L\right)\left(2M+1\right)+JM^{2}\left(Q+4M\right))$
Proposed Method	$O(\left(M-L\right)\left(2M+1\right)+JM^{2}\left(Q+2M\right))$

**Table 2 sensors-23-05061-t002:** Matrix decomposition complexity comparison.

Method	Eigen Decomposition	Singular-Value Decomposition
iMUSIC	$O(JM^{3})$	*not required*
WS-TOPS	$O(JM^{3})$	$O(L^{3}\left(J-1\right)J^{\prime })$
AMI	$O(M^{3})$	*not required*
Proposed Method	$O(M^{3})$	*not required*

**Table 3 sensors-23-05061-t003:** Average estimation error for different source types.

Source	Original AMI ${(}^{\circ })$	Proposed Algorithm ${(}^{\circ })$
Car engine	2.9969	2.4972
water fall	0.6235	0.4362
Hovering drone	1.8102	1.3730
Electric fan	2.3723	0.6860
Train engine	1.0628	1.2502

## Data Availability

Not applicable.

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
