# Peer review of "A Coherent Wideband Acoustic Source Localization Using a Uniform Circular Array"

_sensors, 2023, doi:10.3390/s23115061_

Round 1

Reviewer 1 Report

The paper addresses a coherent wideband acoustic source localization with uniform circular array.

 1- The abstract should be modified. There is not enough information about methodology, proposed work, conclusion in this part. I suggest you structure your abstract as presented in:

 2- The introduction started with some information about sound source localization. The introduction section is short, and it should be longer, but with focus on the details. The introduction should be extended to new published papers for recent years. In the introduction should be expressed the better state-of-art of new methods. The new references will also be examined in this part. I would like to see the articles for last and this year in this section.

 3- I cannot see the details of the proposed method in this article. All the proposed methods were shown in section 3, where there is not enough information about the algorithm. I recommend to authors to show the proposed method by a block diagram.

 4- Figure 2 has some datils where had not been explained well in the text of the article. I recommend explaining well of this figure.

 5- Simulation conditions are not well discussed. The proposed approach was illustrated only on some specific simulations, which is not enough to draw a complete and accurate conclusion about the proposed approach.

 6- This method should be compared with more famous methods in sound source localization to determine the superiority of the proposed method. The evaluations in not enough.

 7- Please, do not forget that the clarity and the good structure of an article are important factors in the review decision. Please read the paper carefully (again) and correct it in English.

Moderate editing of English language.

Author Response

Please find the response in the attached file.

Reviewer 2 Report

 The experimental setup in Fig. 5 (B) needs more elaboration and details.

Some of the equations are basic and can be referred (e.g equation 5). So I would recommend removing some of the formulas from the paper.

As suggested on page 10 of the paper, the proposed algorithm outperformed in most cases, not all cases. Please clarify in which circumstances the algorithm is underperformed and how it can be further improved.  

N/A

Author Response

Please find the response in the attached.

Reviewer 3 Report

Please, see the attached PDF file.

Some typos in the document. 

Author Response

(The authors gave the same response as above.)

Round 2

Reviewer 1 Report

The authors corrected the article based on most of the comments and observations. It can be accepted.

Moderate editing of English language